# In Situ Study of the Impact of Aberration-Corrected Electron-Beam Lithography on the Electronic Transport of Suspended Graphene Devices

**DOI:** 10.3390/nano10040666

**Published:** 2020-04-02

**Authors:** Naomi Mizuno, Fernando Camino, Xu Du

**Affiliations:** 1Department of Physics and Astronomy, Stony Brook University, Stony Brook, NY 11794-3800, USA; 2Center for Functional Nanomaterials, Brookhaven National Laboratory, Upton, NY 11973, USA

**Keywords:** graphene devices, e-beam damage, e-beam lithography, aberration corrected

## Abstract

The implementation of aberration-corrected electron beam lithography (AC-EBL) in a 200 keV scanning transmission electron microscope (STEM) is a novel technique that could be used for the fabrication of quantum devices based on 2D atomic crystals with single nanometer critical dimensions, allowing to observe more robust quantum effects. In this work we study electron beam sculpturing of nanostructures on suspended graphene field effect transistors using AC-EBL, focusing on the in situ characterization of the impact of electron beam exposure on device electronic transport quality. When AC-EBL is performed on a graphene channel (local exposure) or on the outside vicinity of a graphene channel (non-local exposure), the charge transport characteristics of graphene can be significantly affected due to charge doping and scattering. While the detrimental effect of non-local exposure can be largely removed by vigorous annealing, local-exposure induced damage is irreversible and cannot be fixed by annealing. We discuss the possible causes of the observed exposure effects. Our results provide guidance to the future development of high-energy electron beam lithography for nanomaterial device fabrication.

## 1. Introduction

Electron beam lithography (EBL) [1] is one of the most important techniques that allows the definition of sub-micron critical dimensions, essential for realizing quantum wires, dots and constrictions which are used for studying quantum phenomena such as confinement [2], interference, exchange statistics [3], etc. The observation of these and other similar quantum effects usually requires sub-kelvin range temperatures to overcome decoherence (or “dephasing”), which destructs interference effects of electron wave functions. For practical reasons, it is highly desirable to push up the upper temperature limits at which quantum devices operate; however, this requires the fabrication of devices with nanometer-range critical dimensions. With traditional low-energy (≤100 keV) EBL reaching its resolution limit at ~10 nm, recently, aberration-corrected EBL (AC-EBL), which uses an AC scanning transmission electron microscope (STEM) as the exposure tool, demonstrated single-nanometer resolution patterning in conventional electron beam resists commonly used for device fabrication [4,5,6]. AC-EBL thus becomes an appealing option for the fabrication of quantum devices with critical dimensions below 10 nm. The use of the high-energy and tightly focused electron beam available in a transmission electron microscope (TEM) [7,8,9,10,11,12,13,14] and STEM [15,16] has been extensively studied as a lithographical tool for the fabrication of diverse nanoscale structures in suspended graphene. In these works, unintentional electron beam exposure-induced damage has been observed and its prevention was identified as a key factor for the successful realization of practical nanostructure fabrications. Despite these developments, the impact of high-energy electron beam exposure on the electronic and charge transport properties of electronic materials, especially in an electric field effect device structure (where a global backgate electrode covers the entirety of the channel material) still lacks systematic study.

This work uses suspended graphene field effect devices (see Figure 1) to explore in situ the effect of 200 keV AC-EBL on the electronic transport properties, focusing on the impact of electron beam exposure on device quality. When AC-EBL is performed on a graphene channel (referred as local exposure in this work) or on the outside vicinity of a graphene channel (termed non-local exposure), the charge transport characteristics of graphene can be significantly affected due to charge doping and scattering. While the induced detrimental effect of non-local exposure can be largely reduced by vigorous annealing, local-exposure induced damage is irreversible and cannot be fixed by annealing. We attribute the impact of non-local exposure to the generation of high-angle secondary and backscattered electrons produced when the high-energy incident beam interacts with the global backgate of the device, mainly inducing non-uniform charge doping and Coulomb scattering. Local exposure, on the other hand, causes irreversible structural damage to the graphene lattice, which drastically reduces the quality of the field effect devices.

## 2. Materials and Methods 

Fabrication of suspended graphene devices on SiO_2_/Si is based on a wet-etching method reported previously [17], using exfoliated graphene from highly oriented pyrolytic graphite (HOPG). In a typical device (e.g., Figure 1c), the graphene channel is divided into ~3–6 segments by Au (30 nm)/Cr (2 nm) electrodes. The SiO_2_ underneath graphene is chemically etched, allowing the whole graphene flake to be suspended by the electrodes over the Si backgate.

To in situ monitor the charge transport characteristics of the devices, a home-modified STEM insert with four current/voltage electrodes and one gate electrode was prepared based on a commercial one-electrode bias applying holder (Hitachi, Tokyo, Japan, Figure 1a). With the holder, in situ electrical measurements in the STEM were implemented in the configuration shown schematically in Figure 1b. A LabVIEW-based computer program controls a Keithley 2636A dual-channel source/meter unit for the acquisition of resistance vs. backgate voltage (R vs. V_BG_) curves, and for the implementation of in situ Joule-heating (current annealing) protocols which are presented in the next section. AC-EBL was implemented in a Hitachi HD 2700C aberration-corrected scanning transmission electron microscope (Hitachi, Tokyo, Japan) equipped with a pattern generator from JC Nabity Lithography Systems (NPGS, Bozeman, MT, USA). Due to the presence of the thick Si backgate, sample imaging in the STEM could not be performed with the conventional high-angle annular dark field detector (HAADF). Instead, a secondary electron detector on the side of the incoming beam was used to focus the beam and to image the sample. The beam energy during exposures was 200 keV with an estimated beam current of 35 and 180 pA at 35 and 80 µm apertures, respectively. AC-EBL on suspended graphene devices mainly consisted of writing a periodic array of single points with an exposure time per point of ~10 s.

Ex situ electrical characterization and current annealing protocols were performed after AC-EBL with the sample inside a cryogenic insert cooled to 80 K. 

## 3. Results

First, we demonstrated that sub-10 nm structures can be directly “sculptured” on suspended graphene. Typical exposure patterns are shown in Figure 1d,e, which were created by directly (without e-beam resist) focusing the 200 keV beam on free-standing graphene (the beam did not impact any structure other than graphene). Depending on the quality of the focused beam profile, antidots can be created following a designed pattern, with diameters down to 2 nm (smaller antidot diameters could be attained with further exposure time optimization). The exposure time for creating a single antidot (~10 s) is significantly longer compared to the conventional electron beam resist approach, but may be improved by using larger beam currents with better-aligned electron beam optics. 

Next, we focused on the impact of electron beam exposure on the electronic properties of graphene electric field effect devices with global backgate electrodes, considering two exposure conditions: non-local and local exposure. In non-local exposure, we studied the potential detrimental impact of high-angle secondary and backscattered electrons which originate when the incident beam interacts with the backgate of the devices. Here we exposed the segment of graphene that is in the outside vicinity (~1 µm away) of the channel whose resistance and backgate-dependence is measured in situ. Figure 2a shows the change of resistance over time during and after non-local exposure in device A. When over scanning single frames (~100 × 100 nm) at 0.5 frame/second using a 35-pA beam current for 10 s, a rapidly increasing channel resistance was observed. After the beam was blanked, the channel resistance gradually decreased back towards its pre-exposure value (solid red curve in Figure 2a). Similar behavior was also observed in a point-exposure outside the graphene channel (solid black curve). Here the resistance increase during the non-local exposure was slower after a rapid small initial increase. Once the beam was blanked, the resistance decreased towards the pre-exposure value at a similar rate as the single-frame non-local exposure. 

To further understand the non-local exposure effect on graphene, we measured the backgate-dependence of the resistance of a graphene channel in a different device (device B) before and after non-local exposure. In this measurement (solid black curve in Figure 2b), we first annealed the graphene channel by Joule-heating with a large current (~0.4 mA/µm). This allowed the contaminant/residue on the channel to be largely removed, as indicated by the small neutral point backgate voltage. Here the charge neutral point was located at a backgate voltage of ~ −2V, which corresponds to an impurity doping of just ~4 × 10^10^ cm^−2^. We note that the seemingly broad gate-dependence of resistance is mainly a result of the small vacuum dielectric constant in our suspended structure. Based on the gating curves, the maximum mobility of the devices discussed in this work can be estimated to have a lower bound of ~5000 cm^2^/Vs, which is limited by contact resistance, thermal carrier excitations and phonon scattering at room temperature.

After non-local exposure (solid red curve), the channel became strongly electron-doped, with the charge neutral point shifted to a very large negative backgate voltage, which is no longer measurable in our device (at large backgate voltages the suspended graphene channel collapses due to electrostatic pressure). Such heavy doping from non-local exposure, however, was found to be reversible after thorough Joule-heating as shown by the solid blue curve in Figure 2b. After applying a large current through the channel, the charge neutrality point moved back to the original backgate voltage, and the overall backgate-dependence of the channel resistance was also recovered.

The second type of e-beam exposure we studied was the direct exposure of the graphene channel, which is electrically measured. Here we focused on the extensive direct exposure which is used to sculpture graphene, and aimed to study the impact of such exposure on the un-exposed area in the vicinity (~1 µm) of the exposed points. We started by measuring the as-fabricated gating curve of a channel in device C, which showed rather strong electron doping (black curve in Figure 3a). The channel was then subjected to a large current Joule-heating annealing, which largely removed contaminants/residues and shifted the charge neutrality point to the vicinity of zero backgate voltage (red curve). The sample was then treated with non-local exposure, with a resulting gating-curve (blue curve) indicating strong electron-doping, consistent with what was discussed before for device B. Finally, the channel of the device was directly exposed by the 200 keV beam, which patterned a 30-nm-period triangular array of ~100 points, each exposed with 35 pA beam current for 15 s. After such local exposure, the channel showed significant increased resistance and its gate-tunability was almost completely lost (magenta curve in Figure 3a). The sample was then taken out of the STEM and loaded into a cryogenic insert for measurements at liquid nitrogen temperature (Figure 3b). During this transfer process the sample was unavoidably exposed to the ambient, and its resistance dropped significantly from >20 kΩ to <2 kΩ. We were not able to monitor and track how such resistance drop happened. But the observation here may be related to very strong doping as a result of saturation of the e-beam-induced dangling bonds from ambient molecules. A few in situ Joule-heatings were performed, which did not noticeably improve the quality of the channel: while the overall channel resistance shifted up and down over different Joule-heating current annealings, the pre-exposure gate dependence (hence the transconductance and the field effect mobility of the device) was never recovered (Figure 3b). 

The channel in device B was also imaged very briefly, using a secondary electron detector, at the following sequential stages: as-fabricated, Joule-heating annealing, local exposure and post-exposure annealing (Figure 3c, d, e, and f, respectively). In as-fabricated devices, contamination of graphene is largely due to electron beam resist (PMMA) residue from nanofabrication (Figure 3c), which cannot be completely removed without damaging the graphene device. Such contamination is generally present in all graphene devices where graphene is exposed to electron beam resist during the fabrication process. Joule-heating from current annealing removes/redistributes contaminants, resulting in nearly pristine graphene at the center of the channel as shown in Figure 3d. However, near the electrodes the contaminants/residues remain due to the cooling effect from the electrodes, which act as a thermal reservoir. After local exposure, dendrite-like features with strong contrast are observed on the graphene channel, which cannot be removed effectively by annealing.

## 4. Discussion

To understand the impact of electron beam exposure on the transport characteristics in graphene, we consider both charge scattering and doping effects induced by the exposure. Short-range charge scattering can originate from point defects, where carbon atoms are knocked off by the beam’s high-energy electrons [18,19]. Coulomb scattering can be caused by non-uniformly distributed charge centers formed by trapping of low energy secondary electrons on impurities or on damaged graphene lattice sites. Doping effects can take place directly on the graphene channel, or from trapped charges on the backgate [20].

Our results from non-local exposure strongly indicate the significant effect on the graphene resistivity caused by high-angle (with respect to the surface normal) side-scattered and secondary electrons [21], which are generated when the high-energy beam impacts the Si substrate that forms the backgate of the devices (located ~300 nm below the suspended graphene channel). This interaction between electron beam and Si backgate causes a very large secondary electron “cloud” which can be highly non-local and can affect graphene micrometers away from the direct exposure point. The main effect of non-local exposure appears to be electron doping of graphene [20,22]. Since pristine graphene does not trap electrons, the observed doping effect is likely associated with contamination in the graphene layer or in the backgate. In particular, contaminants on graphene are expected to affect e-beam-associated processes. They may act like a blocking layer effectively increasing the necessary dose for e-beam sculpting, as well as decreasing its resolution. They may also absorb low energy secondary electrons and behave like charge trapping centers, doping graphene and inducing Coulomb scattering, lowering the mobility of the devices.

While the energy of the high-angle backscattered electrons can in principle be high enough to cause atomic displacement (knock-on damage) on graphene, our non-local exposure results suggest that such structural damage effect is quite mild, consistent with the low backscattered electron cross section at high scattering angles [21]. (The angle of backscattered electrons which allow them to reach the nearby graphene channel is estimated, based on the device geometry, to be greater than 70°; the rightmost “non-local” beam in Figure 4 depicts this situation.) However, these electrons could be energetic enough to induce the attachment of contaminants on the graphene channel [23], similarly to what was observed by Clark et al. [16]. Even though these effects drastically affect graphene resistivity, the pre-exposure device characteristic can be recovered by annealing. 

Local exposure, on the other hand, causes much more severe damage to the graphene channel. This is evident from the dramatic degradation of the device characteristics, as well as the irreversibility of the exposure effect. We note that even though our local exposure was performed over a large area (~1 µm × 1 µm) and in total over a long period of time (~30 min), the intentionally exposed area consisting of 100 points is very small compared to the total channel area. The drastic degradation in electronic transport suggests significant damage from high-energy electrons which bombard the vicinity of the exposure points. Puster et al. observed no drastic change in the resistivity of graphene nanoribbon devices without backgate [22]. Consequently, these detrimental high-energy electrons are most likely low-angle and high cross section backscattered electrons emerging from the backgate underneath the exposure points (the “local” beam in Figure 4 depicts this situation). Such backscattered electrons can severely damage graphene in the close vicinity of the directly exposed area. Based on our secondary electron imaging, strong local exposure likely creates dangling carbon bonds, which chemically link to contaminants (from the vacuum chamber or device fabrication residues), resulting in significant reduction of the gate-tunability (hence low field-effect mobility).

In summary, we studied the implementation of AC-EBL in a 200 keV STEM on suspended graphene field effect transistors. We carried out in situ characterization of the impact of electron beam exposure on the devices’ electronic transport during local exposure and non-local exposure conditions. While the detrimental effect of non-local exposure can be largely removed by vigorous annealing, local-exposure induced damage is irreversible and cannot be fixed by annealing. We correlated the exposure damage to graphene with the generation of high-energy secondary and backscattered electrons, whose detrimental impact on graphene’s electronic transport characteristics is dependent on the scattering angle. The results from our study here may provide guidance for the future development of high-energy electron beam lithography for nanomaterial device fabrication. 

## Figures and Tables

**Figure 1 nanomaterials-10-00666-f001:**
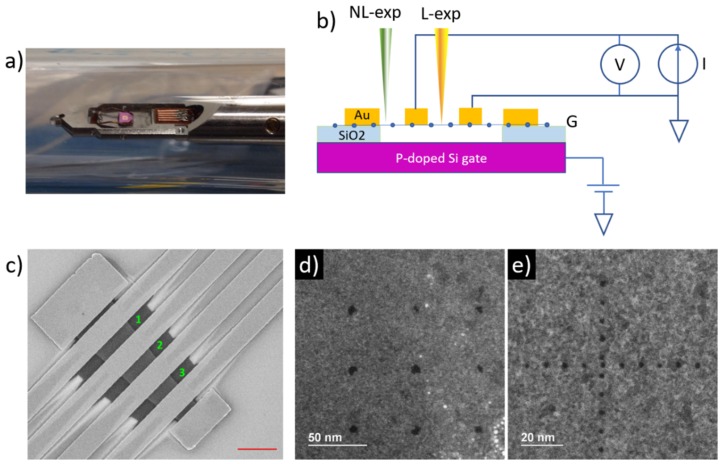
(**a**) Home-modified scanning transmission electron microscope (STEM) holder with five electrodes. (**b**) Schematic showing in situ measurement of a suspended graphene device. The labels “NL-exp” and “L-exp” in the electron beams denote “non-local” and “local” exposure situations, respectively. (**c**) Scanning electron microscope (SEM) micrograph of a finished device. Numbers indicate three electrically independent suspended graphene channels separated by gold electrodes. Scale bar: 2 μm. (**d**,**e**) Sub-10 nm antidots fabricated on suspended graphene via electron beam “sculpturing.” The patterns consist of an array of holes with average diameters of (**d**) 8 nm and (**e**) 2 nm.

**Figure 2 nanomaterials-10-00666-f002:**
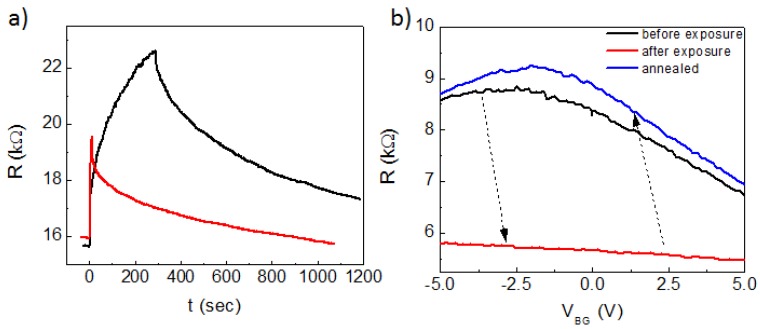
(**a**) Time-evolution of channel resistance during and after non-local exposure. The red curve corresponds to repeated imaging of a 100 x 100 nm window, and the black curve corresponds to a point exposure. The channel resistance in both curves slowly drops back to the pre-exposure values after the beam is blanked. (**b**) Gating curves taken before (black) and after (red) non-local exposure, and after Joule-heating (blue).

**Figure 3 nanomaterials-10-00666-f003:**
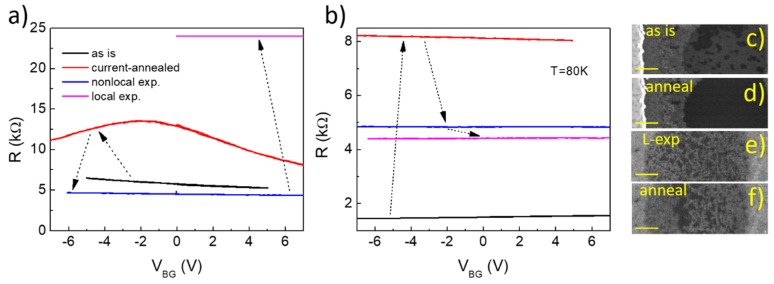
(**a**) Gating curves of a suspended graphene channel during various e-beam exposure steps. (**b**) Gating curves after various Joule-heating annealing attempts. The curves were measured at a sample temperature of 80 K. (**c**–**f**) Secondary electron images of a graphene channel at various sequential stages of e-beam exposure treatments. Scale bar in Figure 3c–f: 100 nm.

**Figure 4 nanomaterials-10-00666-f004:**
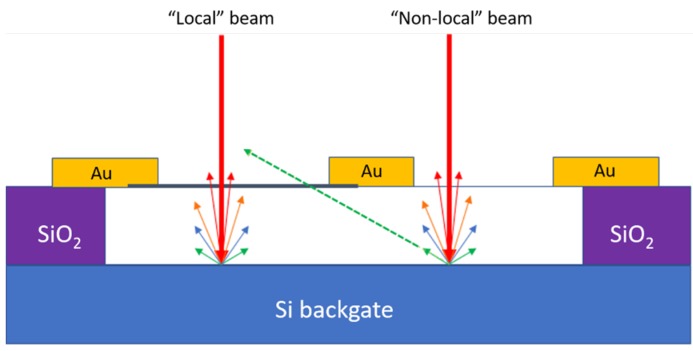
Schematic representation of the role of backscattered electrons on the electrical transport properties of suspended graphene devices. The measured graphene channel is depicted by the black line between the leftmost and the central Au electrodes. In “non-local” exposure (rightmost red beam), only the high-angle (with respect to the Si surface normal) and low cross-section backscattered electrons from the Si backgate manage to reach the graphene channel and mildly affect its resistivity. The dashed green arrow represents this situation. In contrast, in the “local” exposure situation (leftmost beam), low-angle and high cross section backscattered electrons impact the graphene channel, drastically affecting its electrical properties.

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
