# Peer review of "In Situ Study of the Impact of Aberration-Corrected Electron-Beam Lithography on the Electronic Transport of Suspended Graphene Devices"

_nanomaterials, 2020, doi:10.3390/nano10040666_

Round 1

Reviewer 1 Report

The study deals with beam damage effects in graphene on its electrical properties. The setup of the experiment is sound and the conclusions are agreeable. The only comment I have is that the graphene is very dirty as the images show and as all processed graphene is. The impact of the dirt on the electrical properties is huge. Also, the impact of the electron beam and the secondary electrons on the dirt is huge, as figure 3 in the manuscript shows. For my taste, the overall effect of the contamination is underrepresented in the discussion.

Author Response

Response to Reviewer #1

We thank Reviewer#1 for the constructive comments. The Reviewer suggests that we expand the discussion on the effect of processed-related contamination (existing at different degrees on all graphene devices) on graphene’s electronic properties and on the beam-sample interactions: “..the graphene is very dirty as the images show and as all processed graphene is. The impact of the dirt on the electrical properties is huge. Also, the impact of the electron beam and the secondary electrons on the dirt is huge, as figure 3 in the manuscript shows. For my taste, the overall effect of the contamination is underrepresented in the discussion.”

Our response:

In as-fabricated devices, contamination of graphene is largely due to electron beam resist (PMMA) residue from nanofabrication, which cannot be completely removed without damaging the graphene device. Such contamination is generally present in all graphene devices where graphene is exposed to electron beam resist during the fabrication process (e.g., graphene on SiO2, suspended graphene, etc.). The contaminant is indeed expected to affect the e-beam-associated processes. It may act like a blocking layer and effectively increase the necessary dose for e-beam sculpting, as well as decreasing its resolution. It may also absorb low energy secondary electrons and behave like charge trapping centers, and dope graphene and induce Coulomb scattering which lower the mobility of the devices (as discussed in the original manuscript).

Joule heating from current annealing removes/redistributes the contaminants, resulting in nearly pristine graphene at the center of the channel. When exposed to high-energy beam, we observed re-formation of defects inside the graphene channel (Figure 3) which cannot be removed by Joule heating. Such e-beam induced defects may be related to structural damage, or a combination of structural damage and adsorbents. The nature of the e-beam induced defects requires further study.

In terms of electrical properties, we note that the quality of the samples studied here is in fact quite typical. When comparing to graphene on SiO2, one needs to take into account a factor of ~4 difference between the dielectric constants of vacuum and SiO2. Hence the whole gating range of -6V to 6V measured in our work in suspended graphene FETs corresponds to only -1.5V to 1.5V in SiO2-based graphene FETs. Hence the gating curves are in fact quite sharp and the field effect mobility is not low. However, with high-energy e-beam damage from the direct/local exposure, the gating curve becomes almost completely flat which would correspond to a vanishing mobility.

Following the Reviewer’s suggestion, we have added two discussions about process-related contamination in the manuscript. In the last paragraph of the Results section, starting in line 166, we added:

“In as-fabricated devices, contamination of graphene is largely due to electron beam resist (PMMA) residue from nanofabrication (Figure 3c), which cannot be completely removed without damaging the graphene device. Such contamination is generally present in all graphene devices where graphene is exposed to electron beam resist during the fabrication process (e.g., graphene on SiO2, suspended graphene, etc.). Joule heating from current annealing reduces/redistributes contaminants, resulting in nearly pristine graphene at the center of the channel as shown in Figure 3d.”

And, in the third paragraph of the Discussion section, starting in line 196, we added:

“In particular, contaminants on graphene are expected to affect e-beam-associated processes. They may act like a blocking layer effectively increasing the necessary dose for e-beam sculpting, as well as decreasing its resolution. They may also absorb low energy secondary electrons and behave like charge trapping centers, doping graphene and inducing Coulomb scattering, lowering the mobility of the devices.”

Reviewer 2 Report

This paper reports electron beam exposure effects on suspended graphene devices. Recent development on resolution of transmission electron microscope has opened up new possibility for nanometer-scale fabrication of materials with focused electron beam. This is very interesting possibility that can lead not only to realization of novel nanostructures but also to quantum devices working at moderate temperature. Although there are several papers that reports on the electron-beam sculpting, this paper reports, for the first time, details on electron beam irradiation effects onto graphene FET devices. This information is useful for future works to realize quantum devices with the electron-beam sculpting method. I found that the authors have carefully performed experiments and analyzed data. I think this paper can be accepted for publication in Nanomaterials. The followings are comments to the authors for the preparation of revised form of this manuscript.

Just from curiosity, could you tell the limit of resolution of the e-beam sculpting. With aberration-corrected TEM, size of focused electron beam can be ca. 0.1 nm. Is it possible to realize atomic-scale sculpting?

Judging from Fig. 1d, sizes of the holes created in graphene varies. What is the possible reason for this? Dose this mean that atomic/nanometer-scale sculpting is difficult?

I found that mobility of the devices presented in this paper looks very low. Please add values of FET mobility and discuss why the mobilities are low. I guess that amorphous carbon attached on the surface is one of the major reasons. Is it possible to clean whole sample (Joule heating clears only middle part of samples)?

Figure 3c-f need scale bars.

Author Response

Response to Reviewer #2

We thank Reviewer#2 for the constructive comments. 1) The Reviewer asks two questions related to direct-sculpting of holes via aberration-corrected electron-beam. One question is about the resolution limit of the technique and the other about the size-control and reproducibility of patterned features.

Our response:

These are important questions which require extensive research to provide adequate answers. Ideally, one could think that direct e-beam sculpting has the best chance to achieve the highest patterning resolution, since the technique does not require intermediate processes (resist spin casting, resist development, and pattern transfer via etching, for example). In principle, close-to-atomic resolution should be achievable with AC-STEM sculpting, given the sub-nm beam spot size. However, our current tool-related (with a 10-year old, first-generation AC-STEM) experimental conditions might be a limiting factor to achieve the highest resolution possible. We have found, consistent with previous studies, that direct EBL sculpting is a much slower process than conventional resist-based EBL (as mentioned in the first paragraph of the Results section).  The lack of beam stability in our AC-STEM system during these long exposure times (~ 10 s per hole), together with the corresponding loss of optimal beam focusing conditions, can be a limiting factor in achieving sub-nanometer resolution and pattern reproducibility. However, newer generation of AC-STEMs with more stable beam currents and programmable automatic functions (beam current monitoring and automatic focusing routines) could push the technique to sub-nanometer resolution in realistic devices.

Aside from nano-sculpting, AC-EBL using conventional resists and pattern transfer techniques is another promising technique for achieving ultra-high resolutions. Ref. 4 of the original manuscript demonstrated single-digit nanometer resolution using PMMA and HSQ resists, and Ref. 6 reported line edge roughness (LER) values of 1nm using reactive ion etching as the pattern transfer process. These processes could be further optimized using compatible process development protocols followed in the semiconductor industry. Moreover, the use of AC-EBL on conventional resists using secondary electron (SE) mode imaging (not in transmission mode) allows to fabricate FET devices with global gates like those presented in this manuscript, expanding the applicability of the AC-EBL technique. Zhu et al. (Nat. Mater. 8, 808 (2009)) demonstrated atomic resolution imaging using a SE detector, and low energy AC-STEMs (like the 60 keV Nion UltraSTEM™ 100) can achieve atomic resolution below the knock-on threshold damage in graphene. These factors give us hope that AC-EBL technique can be used to fabricate practical quantum devices following conventional (but optimized) device fabrication methods.

2) The Reviewer recommends adding values of the FET mobility and discussing the reasons for the low mobility. The reviewer also asks if there exists a way to clean the totality of the channel: “I found that mobility of the devices presented in this paper looks very low. Please add values of FET mobility and discuss why the mobilities are low. I guess that amorphous carbon attached on the surface is one of the major reasons. Is it possible to clean whole sample (Joule heating clears only middle part of samples)?”

Our response:

In terms of electrical properties, we note that the quality of the samples studied here is in fact quite typical. When comparing to graphene on SiO2, one needs to take into account a factor of ~4 difference between the dielectric constants of vacuum and SiO2. Hence the whole gating range of -6V to 6V measured in our work in suspended graphene FETs corresponds to only -1.5V to 1.5V in SiO2-based graphene FETs. At room temperature, thermal fluctuations would contribute a smearing of ~25 meV near the charge neutral point, which corresponds to a gate broadening of ~3 V in our suspended graphene FETs. In addition, the gating curves were performed in a two-terminal resistance configuration, which includes contact resistance; and electron-flexural phonon scattering can contribute significantly to the resistivity at room temperature in suspended graphene. Considering all these factors, the devices studied here have in fact reasonably sharp gate-dependence, decent quality and low extrinsic defect-associated effects. Following the Reviewer’s suggestion, we have added an estimate of the mobility for the devices studied in this work. Not knowing the contact resistance, we can estimate, based on the gating curves of the total 2-terminal resistance, a lower-bound mobility of ~5000 cm2/Vs for devices.

It would be indeed extremely helpful if one can clean the whole graphene channel. Unfortunately, we are not aware of any post-processing technique which can thoroughly clean the whole sample. Up to now, the only way to achieve over-all clean graphene channels is by hBN-encapsulation, where graphene is never exposed to electron beam resist. For nano-sculpturing, however, the suspended graphene structure is needed here in this work.

We followed the Reviewer’s suggestion and have added the following discussion starting in line 124:  

“We note that the seemingly broad gate-dependence of resistance is mainly a result of the small vacuum dielectric constant in our suspended structure. Based on the gating curves, the maximum mobility of the devices discussed in this work can be estimated to have a lower bound of ~5000 cm2/Vs, which is limited by contact resistance, thermal carrier excitations and phonon scattering at room temperature.”

3) The Reviewer highlights that Figures 3c-f are missing scale bars.

Our response: We have added scale bars to Figures 3c-f and added, in line 169 of the revised manuscript, the corresponding explanatory text: “Scale bar in Figures 3c-f: 100 nm.”

In addition, we took the opportunity to edit a few minor typos, namely (please refer to the revised manuscript):

  • In line 71, we replaced “Figure 2b” by “Figure 1b.”
  • In line 96, we replaced “Figures 2d and 2e” by “Figures 1d and 1e.”